# The Relationship between p-tau217, p-tau231, and p-tau205 in the Human Brain Is Affected by the Cellular Environment and Alzheimer’s Disease Pathology

**DOI:** 10.3390/cells13040331

**Published:** 2024-02-11

**Authors:** Malin Wennström, Nina Schultz, Paula Mille Gallardo, Geidy E. Serrano, Thomas G. Beach, Suchira Bose, Oskar Hansson

**Affiliations:** 1Cognitive Disorder Research Unit, Department of Clinical Sciences Malmö, Lund University, 21428 Malmö, Sweden; nina.schultz@med.lu.se (N.S.); paulamillegallardo@gmail.com (P.M.G.); 2Clinical Memory Research Unit, Department of Clinical Sciences Malmö, Lund University, 21146 Malmö, Sweden; oskar.hansson@med.lu.se; 3Netherlands Institute for Neuroscience, Meibergdreef 47, 1105 BA Amsterdam, The Netherlands; 4Banner Sun Health Research Institute, Sun City, AZ 85351, USA; 5Eli Lilly and Company, Arlington Square West, Bracknell RG12 1PU, UK; bose_suchira@lilly.com; 6Memory Clinic, Skåne University Hospital, 20205 Malmö, Sweden

**Keywords:** tau phosphorylation, co-localization, entorhinal gyrus, inferior temporal gyrus, PART

## Abstract

The levels of p-tau217 and p-tau231 in cerebrospinal fluid (CSF) are associated with early amyloid beta (Aß) changes in the brain, while the CSF levels of p-tau205 are foremost related to tau pathology in the later stages of the disease. To investigate if the three p-tau variants are found to the same degree in different tau structures and if their co-localization is affected by the diagnosis and presence of Aß plaques, we immunostained sections of the entorhinal cortex (EC) and inferior temporal gyrus (ITG) from non-demented controls (NC), patients with Alzheimer’s disease (AD), and primary age-related tauopathy (PART) against p-tau217, p-tau231, and p-tau205 together with Methoxi-X04. An analysis using confocal microscopy showed that the co-localization variable, the Pearson correlation coefficient (PCC), was significantly higher between p-tau231 and p-tau205 in neurofibrillary tangles compared to neuropil threads and dystrophic neurites in plaques. The PCC value between all three p-tau variants in the neuropil threads was significantly lower in the ECs of patients with AD compared to the NC and in the ITGs of patients with AD, with a high Aß load compared to PART. The lowered value was associated with proportionally higher amounts of non-colocalized p-tau231 and p-tau217 compared to p-tau205, and the PCC values were negatively correlated with Aß and the tangle loads in patients with AD, but positively correlated with tangles in PART. These results suggest that the proportion of and co-localization between p-tau217, p-tau231, and p-tau205 are dependent on cellular localization and are altered in response to AD pathology in a spatial–temporal manner.

## 1. Introduction 

Alzheimer’s disease (AD) is characterized by an accumulation and deposition of amyloid beta (Aβ) in the brain. The Aβ depositions vary in size, composition (i.e., content of various extracellular protein components), and morphology, and up to 12 different types have been reported [1,2]. They can be crudely divided into diffuse and cored plaques, where the former are formed by filamentous Aβ and have a loose and more homogenous appearance, while the latter are recognized by their dense cores of fibrillar Aβ and surrounding halos [1]. AD is also neuropathologically characterized by the presence of structures containing aggregated phosphorylated tau (p-tau), which includes neurofibrillary tangles (NFTs), neuropil threads (NTs), and dystrophic neurites. The structures are all neuronal in origin, but the nature of their formation differs. The NFT matures from a pre-tangle, which contains fine diffusely scattered perinuclear abnormal/aggregated fibers of p-tau. During maturation, tau becomes truncated, goes through conformational changes, and aggregates further to form the “basket- or sling-like” shape that is characteristic of an NFT [3]. NTs are instead p-tau-containing fragments of distal parts of axons or dendritic neurites [3], which appear to originate from both NFTs and neuronal cell bodies without NFTs [4]. Finally, dystrophic neurites (DNs) are structures that are foremost found within amyloid plaques. They arise when the neurofilaments, in the presence of amyloid beta depositions, form a ring-like accumulation of neurofilaments. Successively, the neurofilaments continue to aggregate, forming an abnormal neurite containing misprocessed cytoskeletal proteins and p-tau [5]. The p-tau found within the structures are of different types, as tau phosphorylation can occur at several different tau sites (more than 80 sites [6]) and, hence, yield different p-tau variants. Some variants appear to be more harmful than others and cause tau detachment from microtubules, followed by the aggregation and mis-sorting of tau into NFTs, NTs, and dystrophic neurites [7]. Interestingly, studies suggest that some p-tau variants are found to a higher extent in AD compared to other p-tau characteristic neurodegenerative disorders (so-called tauopathies). P-tau217 and p-tau231 are two examples of such variants, and today, both p-tau217 and p-tau231 are considered to be among the earliest biomarkers reflecting pathological changes linked to AD [8,9,10,11,12]. Increased levels of p-tau217 in plasma are associated with both amyloid beta (Aβ) and tau-tangle accumulation [8,13], while a rise in plasma p-tau231 levels is foremost associated with the Aβ plaque load [14,15]. However, it is less clear whether these temporal associations are evident in the brain and if the two p-variants distinguish themselves from each other at a cellular level, and whether this relationship is dependent on Aβ plaques, tau pathology, or diagnosis. In a recent study, we showed that plasma p-tau217 levels correspond well with the amount of p-tau217 in the brain [16]. Moreover, we noted that p-tau217 gave rise to a different staining pattern compared to p-tau231. The p-tau217, and not p-tau231, was found in pre-tangles with granulovacuolar bodies (GVBs), and co-staining against both markers revealed only a partial overlap between p-tau217 and p-tau231 in the NTs and NFTs. This spatial difference in p-tau217 and p-tau231 localization suggests specific, so far unidentified, pathological events in the AD brain. To further study these events and clarify if they are separated from each other and if the consequences differ, we investigate the co-localization between p-tau231 and p-tau217 in NT and NFT as well as in DN. Since we also want to elucidate the spatial–temporal aspect of the co-localization, we additionally include comparisons with a third biomarker, p-tau205. This p-tau variant has lately gained interest, as its levels in the cerebrospinal fluid increase in the later stages of AD [17] and is (unlike most other p-tau variants) related to reduced white matter integrity [18]. P-tau205 is further strongly associated with tau pathology detected with both positron emission tomography (PET) [19] and immunoassays [20], but less with Aβ-PET status [19]. Thus, given its relationship with specific pathological changes, p-tau205 may have the capability to distinguish between different tauopathies. By investigating if the relationship between p-tau217, p-tau231, and p-tau205 is different in patients with AD compared to non-demented controls and patients with primary age-related tauopathy (PART), we hope to narrow down the timing and mechanisms of the specific events implicated in AD-related tauopathy. Such an understanding is crucial for the potential future use of plasma p-tau205, p-tau217, and p-tau231 as monitoring biomarkers in clinical trials and in clinics to distinguish dementia diagnoses. 

## 2. Material and Methods

### 2.1. Individuals Included in the Study: Cohort I and Cohort II

Samples from two cohorts (Cohort I and II) were analyzed in this study. The samples included postmortem-collected brain samples containing entorhinal cortex (EC) (Cohort I and II) and inferior temporal gyrus (ITG) (Cohort II), and plasma collected near the end of life (Cohort II). Cohort I (*n* = 22) consisted of donors from the Netherlands Brain Bank (NBB) and included (*n* = 11) non-demented controls (NC), whereof only *n* = 5 demonstrated tau pathology, and *n* = 11 were patients clinically diagnosed with AD. The demographics of the NC (*n* = 5) and AD (*n* = 11) groups are shown in Table 1. The presence of NFT and NT in Cohort I was scored according to Braak stages I-VI [21], and Aβ plaques were scored as O, A, B, and C, where O = zero, A = some, B = moderate, and C = many [21]. Cohort II included (*n* = 44) participants from an antemortem–postmortem study, the Arizona Study of Aging and Neurodegenerative Disorders and Brain and Body Donation Program at Banner Sun Health Research Institute [22]. The cases in this cohort were scored according to the National Institute on Aging and Reagan Institute (NIA-RI) criteria [23], based on NFT Braak stages (I-VI) [21], and the Consortium to Establish a Registry for Alzheimer’s Disease (CERAD) neuritic Aβ-plaque scores [24], as described earlier [8,13]. The cohort consisted of (*n* = 12) individuals with primary age-related tauopathy (PART) (Braak III-IV, zero-sparse neuritic plaques), (*n* = 21) individuals with an intermediate likelihood of Alzheimer’s disease (Braak stage III-IV, moderate-frequent neuritic plaques) (AD_int_), and (*n* = 11) individuals with a high likelihood of developing Alzheimer’s disease (Braak stage V-VI, frequent neuritic Aβ-plaques) (AD_high_). Demographics of the three groups are shown in Table 1. Procedures for the collection of plasma and brain tissue from Cohort II have been described earlier [8,13]. In addition, Aβ plaque and NFT/NT load (visualized by Campbell-Switzer and Gallyas silver staining, respectively) in the entorhinal cortex, hippocampus, temporal lobe, parietal lobe, and frontal lobe of Cohort II was scored (0–3 per region) and summarized as total plaque load and NFT/NT load (0–15) (for detailed procedures, see [8]). Informed consent for the use of brain tissue, plasma, and clinical data for research purposes was obtained from all subjects or their legal representatives in accordance with the International Declaration of Helsinki [25]. The tissue collection protocols were approved by the medical ethics committee of VU Medical Center in Amsterdam, the Netherlands (Cohort I), and by the Western Institutional Review Board of Puyallup, Washington (USA) (Cohort II). The Swedish Ethical Review Authority approved this study. 

### 2.2. Brain Sample Preparation 

Directly after the autopsy, the brain samples were incubated in paraformaldehyde (PFA) (4%) for either 14–20 h (Cohort I) or 36–72 h (Cohort II). They were thereafter incubated in either phosphate-buffered saline (PBS) with 30% sucrose (Cohort I) or in 2% dimethyl sulfoxide/20% glycerol (Cohort II). The tissue was then sectioned in 40 μm free-floating sections and stored in cryoprotectant at −20 °C (Cohort I) or at room temperature (Cohort II) until used for immunostaining.

### 2.3. Immunostaining

For analysis of co-localization between the p-tau variants, EC sections from patients with AD in Cohort I were stained with antibodies directed against p-tau217, p-tau231, and p-tau205 as well as together with Methoxi-X04 (a fluorescent dye that crosses the blood–brain barrier and binds to fibrillar β-sheet deposits found in Aβ plaques) (detailed information of the antibodies and the dye can be found in Table 2). The EC and ITG sections in Cohort II were only stained against p-tau217, p-tau231, and p-tau205. The sections were incubated for 1 h with blocking solution (BS) containing 5% goat serum (Jackson Immunoresearch, PA, USA) and 0.25% Triton X in KPBS and then incubated with mouse-anti-p-tau231 (ADx Neuroscience, Gent, Belgium) and rabbit-anti-p-tau 205 (Thermo Fischer Scientific, MA, USA) in BS overnight (ON) at 4 °C. The next day, the sections were washed and incubated with secondary antibodies (Alexa 488 goat-anti-mouse and Cy5^TM^ goat-anti-rabbit (Thermo Fischer Scientific)) in BS for 2 h at room temperature (RT). Thereafter, the sections were incubated ON at 4 °C with biotin conjugated rabbit-anti-p-tau217 (IBA493-biotin, Eli Lilly, IN, USA). The following day, the sections were incubated in streptavidin 549 (Thermo Fischer Scientific) in KPBS+ for 1 h. The sections from Cohort I were incubated with bioligand Methoxi-X04 (TOCRIS, Biotechne, Bristol, GB) in KPBS for 1 h at RT. All stained sections were incubated in Sudan Black (1% in 70% ethanol) (Sigma-Aldrich, MA, USA) for 5 min before they were mounted with mounting medium containing DABCO (1,4-diazabicyclo[2.2.2]octane). Out of the eleven stained control cases in Cohort I, only five cases contained NT, and three contained NFTs, while all AD cases in the Cohort I and all PART and AD cases in Cohort II contained, as expected, all types of tau structures.

### 2.4. Analysis of Co-Localization between p-Tau Variants

Co-localization between stained markers was analyzed using confocal microscopy (Zeiss LCM 800, Oberkuchen, Germany). For this purpose, the area of interest (EC in Cohort I and II and ITG in Cohort II) was located with the 20× objective based on area-characteristic landmarks. Pictures (150 × 150 μm) within each brain area were captured with the 63× immersion objective. All images were acquired by employing identical intensity control settings throughout. A total of five images depicting neuropil threads (NTs), three-five images illustrating neurofibrillary tangles (NFT), five images of tau structures within plaques without a core (Tau-P) (exclusively from AD in Cohort I EC), and five images of tau structures within cored plaques (Tau-cP) (exclusively from AD in Cohort I and AD_int_ and AD_high_ in Cohort II) were captured for each individual in each of the previously mentioned regions. The captured images were analyzed by employing the co-localization tools incorporated within the ZEISS ZEN Blue software 2.6. In the case of the NT analysis, the entire acquired image was utilized, whereas for the NFT assessments, a specific region of interest (ROI) delineating the outer border of the NFT was utilized. In Cohort I, tau structures within amyloid plaques without a core (Tau-P) and with a core (Tau-cP) were assessed by delineating the outer border of the plaques. Representative images demonstrating the ROI of the different tau structures as well as staining against the p-tau217, p-tau231, and p-tau205 are found in Figure 1. In Cohort II, dystrophic neurites (DNs) were assessed by delineating the outer borders of the clustered DNs. Differences in co-localization between groups were analyzed by comparing the Pearson correlation coefficient (PCC) for each two-channel comparison. The PCC value quantifies the degree to which two channels follow a simple linear relationship of intensity, where the values range from −1 (an inverse or “anti-co-localization” relationship) to 0 (a random cloud of no relationship) or +1 (a perfect linear slope) [26]. Differences in co-localized and non-co-localized p-tau proportions between diagnoses were analyzed by comparing the total scaled area of pixel count of non-co-localizing p-tau217, p-tau231, and p-tau205 as well as co-localized p-tau217/p-tau231, p-tau217/p-tau205, and p-tau231/p-tau205.

### 2.5. Statistical Analysis

Statistical analysis was performed using SPSS software (version 24 for Mac, SPSS Inc., Chicago, IL, USA). Normal distribution was analyzed using Kolmogorov–Smirnov test. Differences between groups in Cohort I and Cohort II were analyzed using either Mann–Whitney test (when two groups were compared) or Kruskal–Wallis test (when three or four groups were compared). ANCOVA was used when comparisons between diagnoses were adjusted for sex. Correlation analysis was performed using Spearman correlation test, and partial correlation test was used when correlations were adjusted for sex. Results are presented as median, and a value of *p* < 0.05 was considered statistically significant.

## 3. Results and Discussion

### 3.1. Co-Localizations between p-Tau Variants Differ between Tau Structures

Since the fibrillar structure of Aβ and the composition varies between plaque types [27,28], we found it interesting to investigate if the co-localization (the PCC value) of p-tau217, p-tau231, and p-tau205 differed between dystrophic neurites in plaques without a core (DN_C−_) and cored plaques (DN_C+_). An analysis of the EC of AD cases in Cohort I showed that there was no difference in the PCC value of p-tau217/p-tau231, p-tau217/p-tau205, or p-tau231/p-tau205 between DN_C−_ and DN_C+_ (Figure 2A–C). The results suggest that the composition of plaques has a minor effect on the site-specific phosphorylation of tau in DN localized within plaques, and therefore, in the subsequent Cohort II analysis, the DNs in both plaque types were analyzed together.

A further analysis of the p-tau co-localizations in NFT and NT in the EC of Cohort I showed that the PCC value of p-tau217/p-tau205 in NFT was significantly higher compared to those of NT, DN_C−_, and DN_C+_ (*p* = 0.005, *p* = 0.002, and *p* = 0.003, respectively) (Figure 2B), meaning that the co-localization of p-tau217 and p-tau205 is higher in the NFTs compared to the other structures. Similar results were found when analyzing the p-tau variants in the EC of Cohort II; the PCC value of p-tau217/p-tau205 in the NFT was significantly higher compared to that of the NT and DN (*p* = 0.04 and *p* = 0.05, respectively). In addition, a higher PCC value for p-tau231/p-tau205 in the NTs compared to both DN (*p* = 0.05) and NFT (*p* = 0.03) (Figure 2F) was noted. These results indicate that the phosphorylation of tau sites in cell bodies and proximal dendrites (i.e., NFT) differs from distal axons and terminals (i.e., NT) and neurites associated with amyloid plaques (i.e., DN). Such neuronal domain-specific phosphorylation has, to our knowledge, not yet been described, but it is interesting that, for example, aggregates in the perikarya (NFT) are larger than those formed in axons or dendrites (DNs and NTs) [4]. Given this finding, together with the fact that NFT, NT, and DN are formed in distinct and separate ways [3,4], we propose that the alterations in p-tau co-localization are a result of site-specific tau phosphorylation in response to extra- or intra-cellular changes in the three neuronal domains (i.e., perikarya, axon terminal, and dendrites). 

The co-localization between p-tau217/p-tau205 and p-tau231/p-tau205 did not differ between structures in the ITG of Cohort II, but a significantly higher PCC for p-tau217/p-tau231 was found in NFT compared to NT and DN (*p* = 0.004 and *p* = 0.005, respectively) (Figure 2G–I). This result could be due to the fact that ITG is a brain area affected by AD pathology later than EC in terms of the spatiotemporal spread of tauopathy. Hence, an analysis of ITG may reveal differences between the phosphorylation of early sites (p-tau217 and p-tau231) but may fail to detect alterations related to phosphorylation at sites associated with later stages of the disease (p-tau 205).

The PCC values in the different tau structures did not differ between males and females in Cohort I. However, in Cohort II, the females generally showed higher PCC values compared to the males, and the difference was significant in NT between p-tau217/p-tau231 and p-tau217/p-tau205 (*p* = 0.029 and *p* = 0.035, respectively) and in NFT between p-tau231 and p-tau205 (*p* = 0.020) in EC. A significantly higher PCC value was also detected in the females in ITG between p-tau217 and p-tau205 (*p* = 0.046) in DN and between p-tau217 and p-tau205 (*p* = 0.019). Although the result could indicate a sex-dependent impact on the relationship between the p-tau variants, it is important to highlight that the groups, after being divided into males and females, become very small, thereby increasing the risk for a statistical type 2 error.

### 3.2. Co-Localizations between p-Tau Variants in NTs Are Altered in AD Patients

To further investigate if the co-localizations of the p-tau variants are affected by AD pathology, we compared the p-tau PCC values between diagnosis groups. In Cohort I, all three PCC values, i.e., p-tau217/p-tau231, p-tau217/p-tau205, and p tau231/p-tau205, of the NTs were significantly reduced in the AD group compared to the NC group (*p* = 0.02, *p* = 0.04 and *p* = 0.02, respectively) (Figure 3B). Similar results were obtained when analyzing the NTs in the EC of Cohort II; the co-localization between p-tau217/p-tau231 and p-tau231/p-tau205 was significantly lower, and p-tau217/p-tau205 showed a trend to lower co-localization in AD_high_ compared to PART (*p* = 0.02, *p* = 0.04, and *p* = 0.06, respectively). This suggests that AD pathology affects the relationship and co-localization between p-tau variants and that the severity of the AD pathology plays a critical role. This idea was further supported by the results obtained when analyzing the ITG of Cohort II, where the PCC values of p-tau217/p-tau205 and p-tau231/p-tau205 in NTs were, similarly to the EC of Cohorts I and II, lowered in AD_high_ compared to both PART (*p* = 0.003 and *p* = 0.03, respectively) and AD_int_ (*p* = 0.03 and *p* = 0.03, respectively), and the PCC value of p-tau217/p-tau231 in AD_high_ was lowered compared to that of PART (*p* = 0.03) (Figure 3E). In addition, both the p-tau217/p-tau205 and p-tau231/p-tau205 PCC values were significantly lower in the NFTs in AD_high_ compared to those of AD_int_ (*p* = 0.02 and *p* = 0.008, respectively) (Figure 3D). The additional significant differences between AD_high_ and AD_int,_ in ITG, again point towards an effect of the spatiotemporal spreading of tauopathy, where ITG lies downstream of EC and is hence affected less in cases with AD_int_ compared to AD_high._ Since we observed a potential sex-dependent impact on the colocalization between p-tau variants, we also analyzed the PCC values after adjusting for sex. In Cohort I, the PCC value of p-tau217/p-tau205 in the NTs remained significantly lower in AD_high_ compared to PART (*p* = 0.038), and the PCC values of both p-tau217/p-tau205 and p-tau231/p-tau205 in NTs in ITG were still significantly lower in AD_high_ compared to that of PART (*p* = 0.002 and *p* = 0.009, respectively) in Cohort II. Hence, even after adjusting for sex, it seems like the relationship between the two “early” p-tau variants, p-tau217 and p-tau 231, and the “late” ptau205 is affected in AD. 

### 3.3. AD Pathology Alters the Proportions of the p-Tau Variants

The reduced PCC value should be interpreted from the perspective that the PCC value corresponds to the linear relationship between two p-tau intensities in pixels within the selected area. This means that even if the amount of the different p-tau variants increases (as in the case of AD compared to NC), the PCC value does not change as long as the three p-tau variants increase proportionally and the ratio between the area of co-localized and non-colocalized p-tau remains the same. Hence, the reduction in PCC values found in the NTs of AD and AD_high_ suggests that there is a shift in the proportion of co-localizing and non-co-localizing p-tau variants. To verify this, we compared the area fraction of co-localized and non-co-localized p-tau217 vs. p-tau231, p-tau217 vs. p-tau205, and p-tau231 vs. p-tau205 in the NT images. As expected (due to the increased amount of NT in AD patients), the fraction area of co-localized p-tau217/p-tau231 and p-tau217/p-tau205 was significantly higher in the EC in the AD group compared to the NC group in Cohort I (*p* = 0.05 and *p* = 0.05, respectively), and there was a trend towards higher a p-tau231/p-tau205 area fraction in AD compared to NC (*p* = 0.07) (Figure 4A–C). None of the co-localized p-tau fraction areas were significantly altered in the EC of Cohort II when comparing AD_int_ or AD_high_ to PART (*p* = 0.67. *p* = 0.16 and *p* = 0.07), but all three were significantly higher in the ITG of AD_high_ compared to PART (*p* = 0.0001, *p* = 0.0005, and *p* = 0.0005, respectively) (Figure 4F–H). Again, these findings were expected, since tauopathy in EC is one of the characteristics of PART, while less tauopathy is expected to be found in the ITG of these patients [29,30]. 

The fraction area of non-co-localizing p-tau217 and p-tau205 in Cohort I was significantly higher in the EC of the AD group compared to that of the NC group (mean *p* = 0.03 and *p* = 0.005, respectively), but the increase in the p-tau231 area fraction did not reach significance (mean *p* = 0.07). Neither the p-tau217, p-tau231, nor p-tau205 p-tau fraction area was significantly altered in the EC when comparing PART, AD_int_, and AD_high_ (mean *p* = 0.52, *p* = 0.42, and *p* = 0.23, respectively), but in ITG, all of them were significantly higher in AD_high_ compared to PART (mean *p* = 0.0007, *p* = 0.002, and *p* = 0.002, respectively), and p-tau217 as well as p-tau231 were significantly higher in AD_high_ compared to AD_int_ (mean *p* = 0.04 and *p* = 0.02, respectively). Of note, both the co-localized and non-co-localized area fractions of the three p-tau variants increased in the AD group compared to the control (Cohort I) and in AD_high_ compared to PART (ITG in Cohort II). Such AD progression-related increase in the p-tau217 and p-tau23 brain load have previously been shown by us and others [16,31,32], and although similar studies on p-tau205 have not been published to date, it has been shown that the brain load of AT8 (which is directed against p-tau202 and 205) is also increased in the later stages of AD [32]. Whether the biofluid levels (i.e., CSF and plasma) and the brain load of these three p-tau variants increase simultaneously is still under investigation, but associations between the CSF levels of the three p-tau variants and tau-PET have been demonstrated [19,33,34]. The plasma levels of ptau231 also correlate with tau-PET [33], and we have recently shown a strong association between the plasma levels and brain load of p-tau217 [16].

The fraction area of both non-co-localized p-tau217 and p-tau231 also increased to a higher extent compared to p-tau205 in the EC of patients with AD compared to the NC and in ITG of AD_high_ compared to PART and AD_int_. In addition, non-co-localizing p-tau217 and p-tau231 increased proportionally, and the fraction area and PCC value of co-localizing p-tau217/p-tau231 was higher compared to the fraction area and PCC values of co-localization with p-tau205. These results suggest that p-tau217 and p-tau231 increase more or less simultaneously, and to a higher degree at the same location, which is consistent with the literature reporting that they both rise in the early stages of the disease [14]. The p-tau205 is instead associated with later stages of AD progression, and thus, it increases with a delay compared to the other two p-tau variants. Hence, together with the spatio-temporal aspects noted when analyzing the differences in the PCC values between tau structures and diagnosis, we find it likely that the shift in co-localizing and non-co-localizing p-tau proportions occurs along with disease progression. Although there was a tendency for there to be a higher co-localized area fraction of all three p-tau variants in the NFTs in AD_high_ compared to PART, as well as a tendency for there to be a higher p-tau205 area fraction in AD_high_ compared to PART and AD_int_, none of the differences between the co-localized area fraction or non-co-localized p-tau variants reached significance (Appendix A). 

### 3.4. Analysis of Correlations between p-Tau, Aß, and NFT Scores

To further study the relationship between the alterations in the co-localization and proportions of the p-tau variants and severity of AD pathology, we investigated whether the PCC of the three p-tau co-localizations within the tau structures correlated with the neuropathological scores of the Aß plaque load and Braak NFT in the PART and AD (AD_int_ + AD_high_) cases of Cohort II. When analyzing the AD cases in the EC, only the PCC value of the NTs of p-tau217/p-tau231 and p-tau217/p-tau205 correlated negatively with the total NFT (NFT-Tot) and NFT in the EC (NFT-EC). A negative correlation was also found between the PCC values of p-tau217/p-tau205 and p-tau231/p-tau205 in the NFT and NFT-EC (Table 3). After adjusting for sex, only the correlation between NFT-Tot and NT p-tau217/p-tau231 remained. 

In ITG, on the other hand, negative correlations were found between all three NT PCC values (i.e., p-tau217/p-tau231, p-tau217/p-tau205, and p-tau231/p-tau205) and total Aß plaque load (Aß-Tot), Aß plaques in the temporal lobe (Aß-T), and total NFT load (NFT-Tot). The PCC of p-tau217/p-tau205 and p-tau231/p-tau205 was also correlated with the NFT in the temporal lobe (NFT-T) (Table 3). The correlations with the NFT-Tot and NFT-T remained significant after adjusting for sex (NFT-Tot vs. p-tau217/p-tau231, p-tau217/p-tau205, and p-tau231/p-tau205; r = −0.476 *, r = −0.481 *, and r = −0.603 **, respectively, and NFT-T vs. p-tau217/p-tau205 and p-tau231/p-tau205; r = −0.511 * and r = −0.603 **, respectively). The correlation between Aß-Tot and Aß-T and p-tau217/p-tau205 and p-tau231/p-tau205 also remained (Aß-Tot vs. p-tau217/p-tau205 and p-tau231/p-tau205; r = −0.633 ** and r = −0.645 **, respectively, and Aß-T vs. p-tau217/p-tau205 and p-tau231/p-tau205; r = −0.645 ** and r = −0.629 **, respectively).

Furthermore, the DN and NFT PCC values of p-tau231/p-tau205 correlated negatively with NFT-Tot and NFT-T (Table 3), while the PCC values of both tau217/p-tau231 and p-tau231/p-tau205 in the NFTs correlated with Aß-T (Table 3). The correlations with DN were lost after adjusting for sex, while the correlations with the NFT values remained (NFT-Tot vs. p-tau231/p-tau205; r = −0.460 *, NFT-Tot vs. p-p-tau231/p-tau205; r = −0.497 * and Aß-T vs. p-tau231/p-tau205; r = −0.456 *). Only the PCC value of p-tau217/p-tau231 in the NFTs in EC was positively correlated with the NFT-Tot and NFT-EC in PART, (r = 0.84, *p* = 0.02 and r = 0.87, *p* = 0.01). The fact that most of the PCC values of the NT in the patients with AD reduced with increased NFT and Aß load suggests that the AD hallmarks contribute to the alterations in p-tau co-localization/proportions. Whether both contribute equally or if one of them is the driving force remains to be investigated, but it is interesting to note that the negative correlation with NFT is not found in PART, a disorder characterized by tauopathy primarily in the middle temporal lobe and an absence of Aß pathology [29,30]. Hence, together with the fact that AD_high_ showed reduced PCC values compared to PART not only in ITG, but also in EC (despite having a similar amount of tauopathy, as shown in Figure 4), it is likely that the presence of Aß plays a major role. This would cohere with the notion that both p-tau271 and p-tau231 are strongly associated with early Aß pathology [8,10,15,33]. Consequently, given that both p-tau231 and p-tau217 increased to a higher extent compared to p-tau205 in AD_high_ (Figure 4), we speculate that the lowered co-localization with p-tau205, a p-tau primarily associated with later stages of tau pathology, is a result of the gradually increased Aß load which may precede tau pathology. The same speculation can be applied to the altered co-localization between p-tau231 and p-tau271, as the former is foremost associated with Aß pathology, while the latter is associated with both Aß and tau pathology [8,10,15,33]. The differences in the correlation pattern (NFT load vs. PCC) and PCC values between PART and AD further support the idea that PART is not a trajectory towards AD [35], but rather a separate pathology affecting a restricted brain area with a slower longitudinal rate.

## 4. Limitations

Like many previous postmortem studies, this study is limited by the size of the two cohorts. In Cohort I, only a few of the NC patients showed tauopathy in the EC, and Cohort II consisted of only 12 cases of PART, which may have affected the comparative and correlation analyses. In addition, the results obtained after analyzing the differences in the PCC values between males and females should be interpreted with care, as the group sizes after division became very small. Hence, we encourage future studies on larger cohorts to confirm our findings. Finally, although previous studies have shown that p-tau231, p-tau217, and p-tau205 can be found in all maturity stages of NFT [36], some tau epitopes may be lost in the later maturity stages (i.e., in ghost tangles). Despite our goal to only analyze mature NFTs, we cannot completely rule out that the reduced PCC values observed in high AD_high_ (where more ghost tangles are expected to be present) may be partially due to the unintentional inclusion of NFTs in later stages of maturity. 

## 5. Conclusions

Our study suggests that AD pathology, particularly the accumulation of Aß plaque (but not the composition of the Aß plaque), induces a shift in the proportion of co-localized and non-colocalized p-tau217/p-tau231/p-tau205, and that this shift is dependent on the cellular environment of the phosphorylated tau. Hence, these results not only highlight the possibility that different neurodegenerative pathologies introduce different phosphorylation events, but also point to the potential relevance of investigating ratios between different p-tau variants in future plasma biomarker studies. A further analysis investigating the relationship between the p-tau217, p-tau231, and p-tau205 levels in brain homogenates and how these levels are related to the corresponding plasma levels would verify this idea and is therefore encouraged.

## Figures and Tables

**Figure 1 cells-13-00331-f001:**
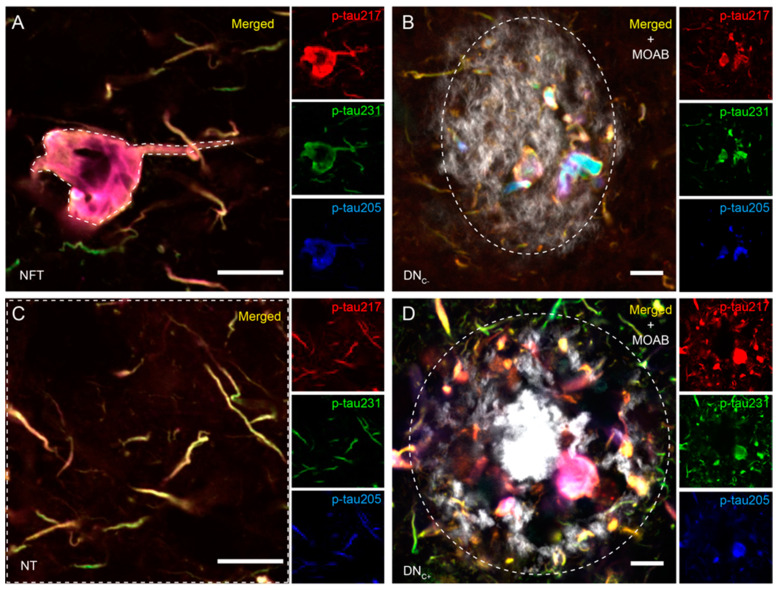
Individual and merged p-tau immunostainings and region of interest (ROI). Images in (**A**–**D**) show the tau structures analyzed in Cohort I. Large images in (**A**–**D**) show p-tau217 (red in the small image), p-tau231 (green in the small image), and p-tau205 (blue in the small image) in neurofibrillary tangles (NFT) (**A**); dystrophic neurites in plaques without a core (DN_C−_) are shown in (**B**); and neuropil threads (NT) and dystrophic neurites in cored plaques are merged (DN_C+_), as shown in (**D**). Representative regions of interest (ROIs) of the different tau structures are indicated by a dashed line in (**A**–**D**). Scalebar = 10 µm (**A**,**C**) and 20 µm (**B**,**D**).

**Figure 2 cells-13-00331-f002:**
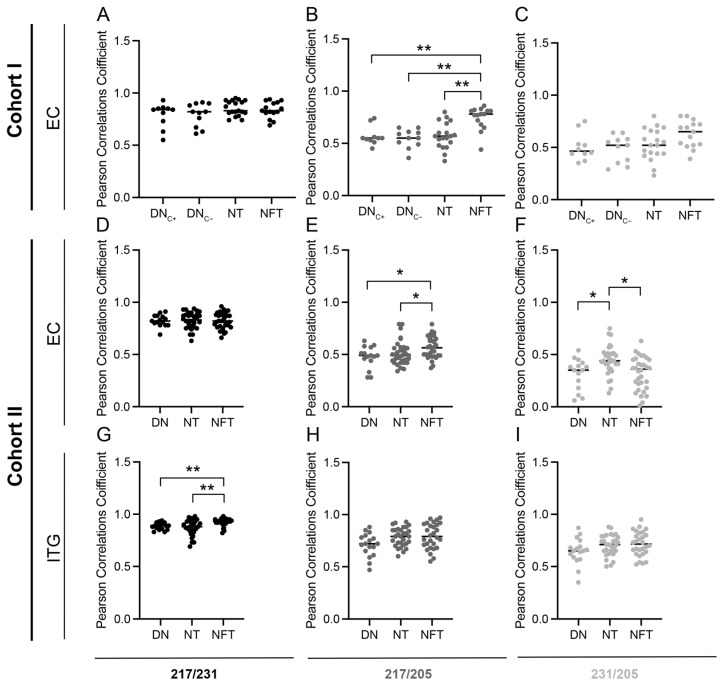
Pearson correlation coefficient (PCC) of p-tau217, p-tau231, and p-tau205 in tau structures. Graphs illustrating the PCC when analyzing the co-localization between p-tau217/p-tau231 (**A**,**D**,**G**), p-tau217/p-tau205 (**B**,**E**,**H**), and p-tau231/p-tau205 (**C**,**F**,**I**) in dystrophic neurites within cored plaques (DN_C+_), dystrophic neurites within plaques without a core (DN_C−_), dystrophic neurites in plaques with and without a core (DNs), neuropil threads (NTs) and neurofibrillary tangles (NFTs) in entorhinal cortex (EC) of Cohort I (**A**–**C**), EC of Cohort II (**D**–**F**), and inferior temporal gyrus (ITG) of Cohort II (**G**–**I**). Data were analyzed using Kruskal–Wallis one-way analysis of variance. Each point represents the mean of five pictures from each individual. * *p* < 0.05; ** *p* < 0.01.

**Figure 3 cells-13-00331-f003:**
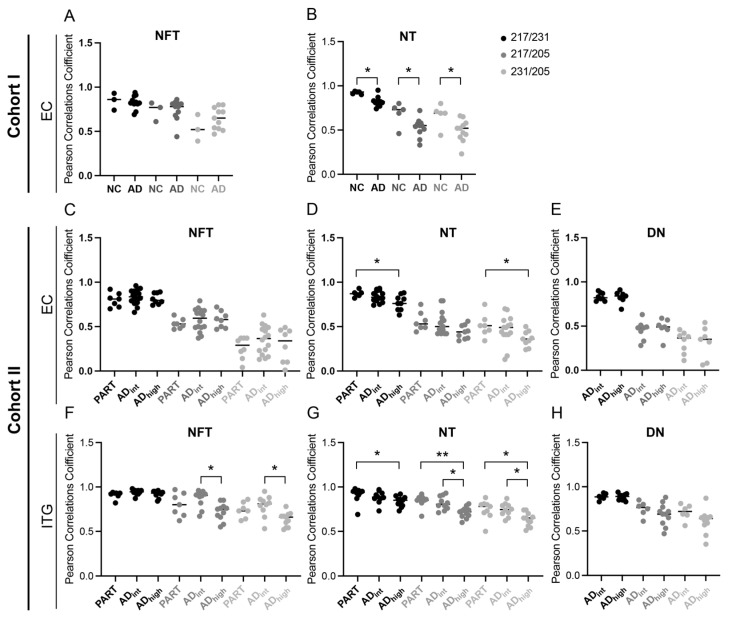
Pearson correlation coefficient (PCC) of p-tau217, p-tau231, and p-tau205 in diagnosis groups. Graphs illustrating the PCC when analyzing the co-localization between the three p-tau variants in neurofibrillary tangles (NFTs) in entorhinal cortex (EC) of Cohort I (**A**), neuropil threads (NTs) in EC of Cohort I (**B**), NFTs in EC of Cohort II (**C**), NTs in EC of Cohort II (**D**), dystrophic neurites (DNs) in EC of Cohort II (**E**), NFTs in inferior temporal gyrus (ITG) of Cohort II (**F**), NTs in ITG of Cohort II (**G**), and DN in EC of Cohort II (**H**). Data were analyzed using Mann–Whitney test (**A**,**B**) and Kruskal–Wallis one-way analysis of variance (**C**–**H**). Each point represents a mean of five pictures from each individual. * *p* < 0.05; ** *p* < 0.01.

**Figure 4 cells-13-00331-f004:**
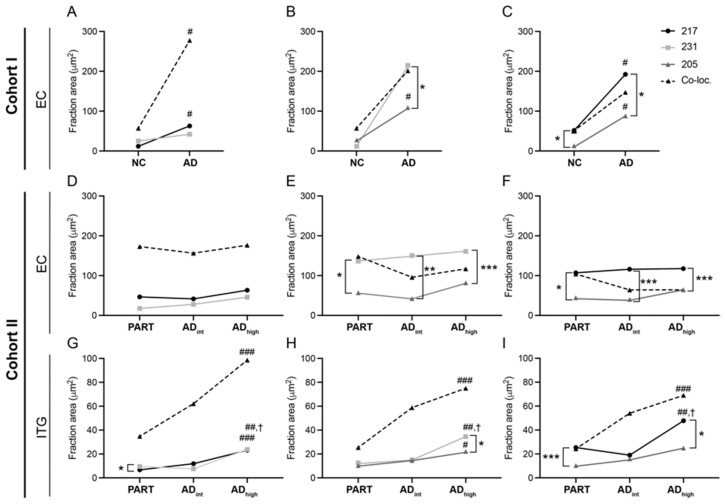
Area fraction of co-localized and non-co-localized p-tau217, p-tau231, and p-tau205 in neuropil threads. Graphs illustrating the area fraction of the three p-tau variants in neuropil threads (NTs) in entorhinal cortex (EC) of Cohort I (**A**–**C**), NT in EC of Cohort II (**D**–**F**), and NT in inferior temporal gyrus (ITG) of Cohort II (**G**–**I**), where p-tau217, p-tau231, and their co-localizations are seen in (**A**,**D**,**G**); p-tau217, p-tau205, and their co-localizations are seen in (**B**,**E**,**H**); and p-tau217, p-tau205, and their co-localizations are seen in (**D**,**F**,**G**). Data were analyzed using Mann–Whitney test (**A**–**C**) and Kruskal–Wallis one-way analysis of variance (**D**–**I**). The points represent the median of the mean area fraction in five images of each case in the different diagnosis groups. * *p* < 0.05, ** *p* < 0.01, and *** *p* < 0.001 when comparing non-co-localized p-tau217 vs. p-tau231, p-tau217 vs. p-tau205, and p-tau231 vs. p-tau205. # *p* < 0.05, ## *p* < 0.01, and ### *p* < 0.001 when comparing PART vs. AD_high_, and † *p* < 0.05 when comparing AD_int_ vs. AD_high_.

**Table 1 cells-13-00331-t001:** Demographic data of individuals included in Cohort I and Cohort II.

Cohort I			Cohort II		
	NC (*n* = 5)	AD (*n* = 11)	PART (*n* = 12)	AD_int_ (*n* = 21)	AD_high_ (*n* = 11)
Age (years)	84 ± 8	82 ± 10	84 ± 9	85 ± 7	83 ± 6
Females (%)	40	45	67	48	9
*APOE4* (%)	0 *	45 †	17	52	73
MMSE (scores)	-	-	25 ± 7	25 ± 9	17 ± 6

AD = Alzheimer’s disease, NC = non-demented control, PART = primary age-related tauopathy. *APOE4* = apolipoprotein E4, MMSE = Mini Mental State Examination. * Missing *APOE4* status from two individuals. † Missing *APOE4* status from three individuals.

**Table 2 cells-13-00331-t002:** Antibodies and dyes used in this study.

Antibody	Antigen	Clone	Specie	Dilution	Incubation Time	Source
	p-tau217	IBA413-biotin	Rabbit	1:500	Overnight	Eli Lilly
	p-tau231	ADx253	Mouse	1:200	Overnight	ADx Neuroscience
	p-tau205	44-738G	Rabbit	1:200	Overnight	Invitrogen
Dye	Target					
	β-sheets	Methoxi-X04	-	1:5000	1 h	TOCRIS

**Table 3 cells-13-00331-t003:** Correlations between PCC values and Aß and NFT scores in Cohort II comprising patients with AD.

**EC**	**DN (*n* = 29)**			**NT (*n* = 29)**			**NFT (*n* = 29)**		
	**217/231**	**217/205**	**231/205**	**217/231**	**217/205**	**231/205**	**217/231**	**217/205**	**231/205**
NFT-Tot	ns	ns	ns	r = −0.649 ***^a^	r = −0.522 **	ns	ns	ns	r = −0.444 *
NFT-EC	ns	ns	ns	r = −0.420 *	r = −0.436 *	ns	ns	ns	r = −0.536 **
**ITG**	**DN (*n* = 29)**			**NT (*n* = 29)**			**NFT (*n* = 29)**		
	**217/231**	**217/205**	**231/205**	**217/231**	**217/205**	**231/205**	**217/231**	**217/205**	**231/205**
Aß-Tot	ns	ns	ns	r = −0.461 *	r = −0.558 *^a^	r = −0.667 **^a^	ns	ns	ns
Aß-T	ns	ns	ns	r = −0.462 *	r = −0.647 **^a^	r = −0.721 ***^a^	r = −0.476 *	ns	r = −0.552 *^a^
NFT-Tot	ns	ns	r = −0.586 *	r = −0.491 *^a^	r = −0.486 *^a^	r = −0.624 **^a^	ns	ns	r = −0.461 *^a^
NFT-T	ns	ns	r = −0.626 *	ns	r = −0.535 *^a^	r = −0.613 **^a^	ns	ns	r = −0.484 *^a^

AD = cases with intermediate and high likelihood of Alzheimer’s disease, EC = entorhinal cortex, ITG = inferior temporal gyrus, DN = dystrophic neurite, NFT = neurofibrillary tangle, NT = neuropil thread, NFT-Tot = total NFT, NFT-EC = NFT in the EC, NFT-ITG = NFT in the ITG, Aß-Tot = total Aß plaque load, Aß-T = Aß plaques in the temporal lobe, and ns = not significant. Data were analyzed using Spearman correlation test. ^a^ indicates values that are significant after adjusting for sex. Data were analyzed using Partial correlation test. * *p* < 0.05, ** *p* < 0.01, and *** *p* < 0.001.

## Data Availability

The authors confirm that the data supporting the findings of this study are available within the article and its Appendix A.

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
