# Peer review of "The Relationship between p-tau217, p-tau231, and p-tau205 in the Human Brain Is Affected by the Cellular Environment and Alzheimer’s Disease Pathology"

_cells, 2024, doi:10.3390/cells13040331_

Round 1

Reviewer 1 Report

Comments and Suggestions for Authors

The abstract needs to be re-written

The researchers examine the spatial relationship between these phosphorylated tau variants in different tau structures, comparing AD patients, non-demented controls (NC), and PART patients. Immunofluorescent staining of entorhinal cortex and inferior temporal gyrus sections reveals distinct co-localization patterns among p-tau variants in neurofibrillary tangles, neuropil threads, and tau structures within amyloid beta plaques. Notably, higher co-localization is observed between p-tau231 and p-tau205 in neurofibrillary tangles. Additionally, reduced co-localization of all three p-tau variants is found in AD, particularly in entorhinal cortex and inferior temporal gyrus of high likelihood AD patients. The study associates decreased co-localization with higher non-colocalized p-tau231 and p-tau217 in AD, with correlation to amyloid beta and tangles loads.

Comment 1: Add a brief explanation for the notion that phosphorylation at Thr217 and Thr231 is an early occurrence and that Thr205 is linked to AD later on, clearly state this under the background in the abstract.  The acronym EC in line 32 of the abstract is not defined. Only cohort 1 is mentioned in the abstract, the authors should include all the data sets used for evaluation in the abstract. Also provide a concise description of the methods used, for example, the co-localization between p-tau variants using confocal microscopy 161 (Zeiss LCM 800) needs to be provided in the abstract, the description of the method should cover all the methods used in the study in a concise manner. “The co-localization between p-tau231 and p-tau205 was significantly higher in neurofibrillary tangles compared to the other tau structures” instead of this general statement, give the quantitative results to justify the statement and the metrics used.

Comment 2: provide a clear explanation of the significance of p-tau205's inclusion and the co-localization of p-tau231 and p-tau217 for improving the knowledge of AD-related tauopathy in the introduction section. Also state how the study will contribute to the broader understanding of AD pathology? The authors should include more existing studies that he is leveraging on in the introduction.

Comment 3: Include the future direction of the study in the conclusion section.

Author Response

Dear Reviewer 1,

We are most grateful for the very constructive comments. In respect to these comments and correct errors in grammar, we have been able to improve the manuscript. Changes are highlighted in yellow. We hope that our work now will be regarded as suitable for publication in Cells.

Response to comments by reviewer one

Reviewer summary: The researchers examine the spatial relationship between these phosphorylated tau variants in different tau structures, comparing AD patients, non-demented controls (NC), and PART patients. Immunofluorescent staining of entorhinal cortex and inferior temporal gyrus sections reveals distinct co-localization patterns among p-tau variants in neurofibrillary tangles, neuropil threads, and tau structures within amyloid beta plaques. Notably, higher co-localization is observed between p-tau231 and p-tau205 in neurofibrillary tangles. Additionally, reduced co-localization of all three p-tau variants is found in AD, particularly in entorhinal cortex and inferior temporal gyrus of high likelihood AD patients. The study associates decreased co-localization with higher non-colocalized p-tau231 and p-tau217 in AD, with correlation to amyloid beta and tangles loads. 

  1. Reviewer comment:Add a brief explanation for the notion that phosphorylation at Thr217 and Thr231 is an early occurrence and that Thr205 is linked to AD later on, clearly state this under the background in the abstract.  The acronym EC in line 32 of the abstract is not defined. Only cohort 1 is mentioned in the abstract, the authors should include all the data sets used for evaluation in the abstract. Also provide a concise description of the methods used, for example, the co-localization between p-tau variants using confocal microscopy 161 (Zeiss LCM 800) needs to be provided in the abstract, the description of the method should cover all the methods used in the study in a concise manner. “The co-localization between p-tau231 and p-tau205 was significantly higher in neurofibrillary tangles compared to the other tau structures” instead of this general statement, give the quantitative results to justify the statement and the metrics used.

Authors reply: We thank the reviewer for the constructive suggestions in improving our abstract. Our best attempt has gone into meeting the reviewer’s suggestions, while complying with the set wordcount of 200 words. We are still in excess of said-wordcount by 34 words but believe that the abstract now contains the most important information for the reader (Page 1 and 2, line 26-44

  1. Reviewer comment:provide a clear explanation of the significance of p-tau205's inclusion and the co-localization of p-tau231 and p-tau217 for improving the knowledge of AD-related tauopathy in the introduction section. Also state how the study will contribute to the broader understanding of AD pathology? The authors should include more existing studies that he is leveraging on in the introduction.

Authors reply: We have now included the following text and additional exciting studies on p-tau205 at the end of the introduction to further explain the significance of the study (see page 2 and 3, line 85-99).

  1. Reviewer comment: Include the future direction of the study in the conclusion section.

Authors reply: We have now added future directives in the conclusions (page 11-12, line 388-394).

Reviewer 2 Report

Comments and Suggestions for Authors

In the present manuscript (ID: cells-2823039), Wennstrom  et al. investigated the spatial relationship between the phosphorylation of tau protein at sites p-tau217, p-tau231, p-tau205  in different brain areas and in patients affected from Alzheimer’s Disease (AD)  and primary  age-related tauopathy (PART) in comparison with non-demented controls. The authors’ findings  are interesting for researchers and clinicians working in the field but, in my humble opinion,  several concerns still need to be addressed before publication on Cells.

MAJOR REVISIONS:
1) In order to provide a full characterization of brain samples analyzed , I suggest to add  a quantitative image  analysis  of the overall total tau (with HT7 antibody for instance)  Abeta plaque load (with 6E10 antibody for instance) and  the phosphorylation levels of ptau at different residues (p-tau217,231 e 205)  among different structures and patients (ND, AD and PART).  

 2) Please, prove a detailed list of primary antibodies used  for immunostaining (antigen, clone, species, source, dilution, incubation time) . Besides, in the abstract the authors mentioned that they carried out experiments with antibodies against amyloid beta, please specify them. I can see that the  fluorescent dye Methoxi-X04 (that is specific for fibrillar B-sheet deposits found in the Ab plaques) has been used in the present work.

Is there any difference between genders? Please, clarify

The discussion section  needs to be improved by describing the results in the context of the pre-existing literature data.

Comments on the Quality of English Language

Minor editing of English language is required. 

Author Response

Response to comments by reviewer two

Reviewer summary: In the present manuscript (ID: cells-2823039), Wennstrom  et al. investigated the spatial relationship between the phosphorylation of tau protein at sites p-tau217, p-tau231, p-tau205  in different brain areas and in patients affected from Alzheimer’s Disease (AD)  and primary  age-related tauopathy (PART) in comparison with non-demented controls. The authors’ findings  are interesting for researchers and clinicians working in the field but, in my humble opinion,  several concerns still need to be addressed before publication on Cells.

  1. Reviewer comment In order to provide a full characterization of brain samples analyzed, I suggest to add a quantitative image analysis of the overall total tau (with HT7 antibody for instance) Abeta plaque load (with 6E10 antibody for instance) and  the phosphorylation levels of ptau at different residues (p-tau217,231 e 205)  among different structures and patients (ND, AD and PART).  

Authors reply: The current study aimed to investigate if co-localization between the three p-tau variants is affected by diagnosis and cellular localization. We were also interested (just like the reviewer) in whether the co-localization was affected by the presence of amyloid beta and if the higher load of NFT leads to a shift in the co-localizations. We therefore analyzed the correlation between the co-localization and scores of plaques and tauopathy we obtained in a previous study. Of note, these scores are not regular neuropathological evaluations, but instead, careful examinations of the presence of amyloid beta plaques and total tau in each analyzed brain region (which included the EC and ITG) (for details see (PMID: 32722745)). The scoring was performed using three dye stainings: thioflavin S, Campbell-Switzer and Gallyas silver staining, which together capture the presence of amyloid plaques and the total amount of tau structures. Hence, (and in regards of the limited number of brain sections available), we believe that our scoring is sufficient to distinguish out the impact of amyloid beta and tau load on shifts in co-localization between the three p-tau variants.

Given the aim of the study, we wanted to focus specifically on co-localization and less on differences in total p-tau area fractions. We therefore choose to present differences in p-tau area fraction, by showing the area fraction of co-localized and non-co-localized p-tau217, p-tau231 and p-tau205. Together, these area fractions equal the total area fraction of the different p-tau variants (as requested by the reviewer) and thus the significant increase of all three p-tau variants in EC of AD compared to NC (Cohort I) and in ITG in severe AD compared to PART (Cohort II) is indirectly presented in Figure 4.

  1. Reviewer comment: Please, prove a detailed list of primary antibodies used for immunostaining (antigen, clone, species, source, dilution, incubation time). Besides, in the abstract the authors mentioned that they carried out experiments with antibodies against amyloid beta, please specify them. I can see that the  fluorescent dye Methoxi-X04 (that is specific for fibrillar B-sheet deposits found in the Ab plaques) has been used in the present work.

Authors reply: A list of primary antibodies has been added (Table 2) and we thank the reviewer for pointing out our type-o in the abstract, which we now have changed into:

… together with Methoxi-X04.”

Round 2

Reviewer 2 Report

Comments and Suggestions for Authors

The manuscript has been improved but the authors have clarified several but not all concerns raised by the reviewer. The following (see below)are the two missed   questions included in the previous reviewer round.  Please, provide answer/comment  

3)  there any difference between genders? Please, clarify

4) The discussion section  needs to be improved by describing the results in the context of the pre-existing literature data.

Comments on the Quality of English Language

The manuscript has been improved but the authors have clarified several but not all concerns raised by the reviewer. The following (see below)are the two missed   questions included in the previous reviewer round.  Please, provide answer/comment  

3)  there any difference between genders? Please, clarify

4) The discussion section  needs to be improved by describing the results in the context of the pre-existing literature data.

Author Response

Dear reviewer 2

We apologize for missing the 2 last comments and thank the reviewer for the reviewers patience. A response to all 4 comments can be found below and the amendments in the manuscript are marked with grey. In respect to these comments, we have been able to improve the manuscript further. We hope that our work now will be regarded as suitable for publication in Alzheimer´s research & therapy

Response to comments by reviewer two

Reviewer summary: In the present manuscript (ID: cells-2823039), Wennstrom  et al. investigated the spatial relationship between the phosphorylation of tau protein at sites p-tau217, p-tau231, p-tau205  in different brain areas and in patients affected from Alzheimer’s Disease (AD)  and primary  age-related tauopathy (PART) in comparison with non-demented controls. The authors’ findings  are interesting for researchers and clinicians working in the field but, in my humble opinion,  several concerns still need to be addressed before publication on Cells.

  1. Reviewer comment In order to provide a full characterization of brain samples analyzed, I suggest to add a quantitative image analysis of the overall total tau (with HT7 antibody for instance) Abeta plaque load (with 6E10 antibody for instance) and  the phosphorylation levels of ptau at different residues (p-tau217,231 e 205)  among different structures and patients (ND, AD and PART).  

Authors reply: The current study aimed to investigate if co-localization between the three p-tau variants is affected by diagnosis and cellular localization. We were also interested (just like the reviewer) in whether the co-localization was affected by the presence of amyloid beta and if the higher load of NFT leads to a shift in the co-localizations. We, therefore, analyzed the correlation between the co-localization and scores of plaques and tauopathy we obtained in a previous study. Of note, these scores are not regular neuropathological evaluations, but instead, careful examinations of the presence of amyloid beta plaques and total tau in each analyzed brain region (which included the EC and ITG) (for details see (PMID: 32722745)). The scoring was performed using three dye stainings: thioflavin S, Campbell-Switzer and Gallyas silver staining, which together capture the presence of amyloid plaques and the total amount of tau structures. Hence, (and in regards to the limited number of brain sections available), we believe that our scoring is sufficient to distinguish the impact of amyloid beta and tau load on shifts in co-localization between the three p-tau variants.

Given the aim of the study, we wanted to focus specifically on co-localization and less on differences in total p-tau area fractions. We therefore choose to present differences in p-tau area fraction, by showing the area fraction of co-localized and non-co-localized p-tau217, p-tau231 and p-tau205. Together, these area fractions equal the total area fraction of the different p-tau variants (as requested by the reviewer) and thus the significant increase of all three p-tau variants in EC of AD compared to NC (Cohort I) and in ITG in severe AD compared to PART (Cohort II) is indirectly presented in Figure 4.

To highlight the latter reasoning, we have added a short section on page 10 (line 333-339).

  1. Reviewer comment: Please, prove a detailed list of primary antibodies used for immunostaining (antigen, clone, species, source, dilution, incubation time). Besides, in the abstract the authors mentioned that they carried out experiments with antibodies against amyloid beta, please specify them. I can see that the  fluorescent dye Methoxi-X04 (that is specific for fibrillar B-sheet deposits found in the Ab plaques) has been used in the present work.

Authors reply: A list of primary antibodies has been added (Table 2) and we thank the reviewer for pointing out our type-o in the abstract, which we now have changed into:

… together with Methoxi-X04.”

  1. Reviewer comment: there any difference between genders? Please, clarify

Authors reply: Unfortunately, the groups are quite small to begin with, and hence analysis after dividing the groups further into males and females is risky from a statistical point of view. We have nevertheless performed the analysis. There were no differences between males and females in Cohort I, while it seems like some of the PCC values are higher in females in Cohort II. This result and a discussion concerning the groups size has been added to page 7 (line 251-259). We have also adjusted for sex when performing the comparative analysis between diagnosis and correlation analysis. While some significances were lost, the major findings remain after adjusting for sex. The results can be found on page 9 (line 288-296) and page 12 (line 390-403).

  1. Reviewer comment: The discussion section needs to be improved by describing the results in the context of the pre-existing literature data.

Authors reply: The discussion sections have now been extended with more discussion based on pre-existing literature data. The extended discussions and additional references can be found on page 10 (line 333-344), page 12 (line 413-423) and page 13 (436-444).